# Localized Insulin-Derived Amyloidosis in Diabetes Mellitus Type 1 Patient: A Case Report

**DOI:** 10.3390/diagnostics13142415

**Published:** 2023-07-20

**Authors:** Jan Hrudka, Eva Sticová, Magdaléna Krbcová, Klára Schwarzmannová

**Affiliations:** 1Department of Pathology, 3rd Faculty of Medicine, Charles University, University Hospital Kralovske Vinohrady, 100 34 Prague, Czech Republic; 2Department of Internal Medicine, 3rd Faculty of Medicine, Charles University, University Hospital Kralovske Vinohrady, 100 34 Prague, Czech Republic; 3Department of Plastic Surgery, 3rd Faculty of Medicine, Charles University, University Hospital Kralovske Vinohrady, 100 34 Prague, Czech Republic

**Keywords:** insulin, amyloidosis, diabetes mellitus, type 1, type 2, injection, glycemia

## Abstract

Localized insulin-derived amyloidosis (LIDA) is a rare local complication of subcutaneous insulin application occurring in patients with diabetes type 1 and 2. A 45-year-old woman with an 11-year history of insulin-dependent diabetes mellitus type 1 underwent a mini-abdominoplasty and excision of a long-standing palpable mass in left hypogastric subcutaneous tissue in the area of long-term insulin application. Histopathological examination revealed insulin amyloidosis as a substrate of the mass lesion. Several months after surgery, there was a transient improvement in previously poor diabetes compensation. In addition to local allergic reactions, abscess formation, scarring, lipoatrophy/dystrophy, and lipohypertrophy, LIDA broadens the differential diagnostic spectrum of local insulin injection complications. LIDA has been described as a cause of poor glycemia compensation, probably due to the conversion of soluble insulin into insoluble amyloid fibrils, which prevents insulin from circulating in the blood and regulating glucose blood concentration. Improvement in diabetes compensation has been described in several reports, including our case. LIDA is a rare local complication of subcutaneous insulin application; accurate diagnosis and treatment have clinical consequences. Immunohistochemical or immunofluorescence distinction from other amyloid types is highly recommended.

## 1. Introduction

Localized insulin-derived amyloidosis (LIDA) is a rare local complication of subcutaneous insulin application. Long-term insulin therapy is essential in patients with type 1 diabetes mellitus (T1DM) and approximately one-third of patients with type 2 diabetes [1].

## 2. Case Description

A 45-year-old woman with a 16-year-long history of insulin-dependent T1DM presented to the department of plastic surgery for a scheduled mini-abdominoplasty. She had a painful, skin-colored, palpable subcutaneous mass in the left hypogastric region, where she usually applied insulin injections. The patient had undergone subcutaneous mass excision in the same area 9 years before the current presentation, which had been histologically diagnosed as insulin-induced lipodystrophy. Apart from T1DM, her medical history was unremarkable. The patient was first diagnosed with T1DM at 30 when she developed hyperglycemia with mild ketoacidosis and tested positive for glutamate decarboxylase and tyrosine phosphatase antibodies. Intensified insulin therapy was initiated with a daily dose of approximately 50 IU (rapid-acting insulin aspart 36 IU and ultralong-acting basal insulin degludec 12 IU per day). In the long term, her compensation has been poor, with glycated hemoglobin (HbA1c) between 80 and 90 mmol/mol. Despite this, the patient was free of macrovascular and microvascular complications, presumably due to her young age. Her urine albumin to creatinine ratio (ACR) was 1.33, and she showed no diabetic foot syndrome or retinopathy.

At the recent presentation, the ultrasonographic examination showed a subcutaneous inhomogeneous hypovascularized mass measuring ca. 14 × 10 × 4 cm^3^, with slight size progression compared to previous ultrasound examinations in the available documentation. The ultrasound finding was regarded as compatible with the previous diagnosis of lipodystrophy. A mini-abdominoplasty was performed with skin excision and subcutis of 135 × 100 × 40 mm without any peri- and postoperative complications. The excision was made followed by standard histopathological examination with formalin fixation. On the cut section upon gross examination, subcutaneous fat tissue with firm, ocher-yellow vaguely defined portions was apparent. The representative tissue blocks were formalin-fixed and paraffin-embedded (FFPE) and processed. The histological and immunohistochemistry findings shown in Figure 1 led to the diagnosis of localized insulin-induced amyloidosis.

Following discharge, the patient was regularly seen by her diabetologist at an interval of three months. Her compensation of T1DM improved six months after surgery with an HbA1c of 60 mmol/mol. One year after the excision, the values returned to the pre-surgery range (81 mmol/mol), but no resistance was found on the site of insulin application.

## 3. Discussion

Amyloidosis refers to the extracellular accumulation of amyloid fibrils in various tissues and organs, which may result in the disruption of their function. Amyloid fibrils are a specific type of protein aggregate that may cause significant illness leading to the patient’s death in the case of systemic involvement. In contrast to systemic amyloidosis, localized amyloidosis manifests as an amyloid tumor causing only local complaints with mild symptoms [2]. According to the nomenclature committee of the International Society of Amyloidosis, the term “amyloid” means mainly extracellular tissue deposits of protein fibrils, recognized by specific histological properties, such as green–yellow birefringence after staining with Congo red [3]. Currently, there are 36 human amyloid proteins, of which 14 appear to only be associated with systemic amyloidosis and 19 appear as localized forms. Three proteins can occur both in localized and systemic amyloidosis. Amyloids consist of an amyloid fibril protein deposited as insoluble fibrils, mainly in the extracellular spaces of organs and tissues [4].

Additionally, in vivo, amyloid fibrils contain a serum amyloid P-component and proteoglycans, mainly heparan sulfate proteoglycan [3]. An amyloid fibril protein occurs in tissue deposits as rigid, non-branching fibrils approximately 10 nm in diameter. The fibrils bind the dye Congo red and exhibit green birefringence when viewed by polarization microscopy.

The common types of acquired amyloidogenic proteins include, i.e., immunoglobulin light chains or heavy chains in plasma cell myeloma, serum-amyloid-associated protein (SAA) in various types of chronic inflammation, transthyretin in senile amyloidosis, β2-microglobulin in systemic amyloidosis in patients undergoing long-term dialysis, and various hormones (atrial natriuretic peptide and calcitonin) in cases of localized amyloidosis [5]. Insulin may compose localized amyloidosis in insulin-producing neuroendocrine tumors of the pancreas (insulinoma). LIDA arising due to long-term diabetes treatment may be labeled iatrogenic. In addition to insulin, amyloidosis may accompany the injection of liraglutide (glucagon-like peptide-1-mimicking drug) [6].

LIDA represents a rare phenomenon occurring in subcutaneous fat tissue after long-term insulin injection following diabetes mellitus treatment, first reported in 1983 [7]. Dische et al. described subcutaneous amyloidosis following porcine insulin application [8]. Firm waxy masses have been generated following subcutaneous insulin application in experimental mice [2]. All patients with type 1 and approximately 30% of patients with type 2 diabetes mellitus undergo long-term subcutaneous insulin injection therapy [1]. To date, several small case series and case reports documenting LIDA following diabetes treatment have been published [9,10,11,12,13,14,15,16,17,18,19,20,21,22,23,24]. The most common LIDA site reported in the literature is the abdomen, with sizes varying from 1 cm to 18 cm. LIDA may occur after 4–47 years of subcutaneous insulin administration [1]. Nagase et al. described poor insulin absorption and subsequent poor hyperglycemia control following injection into LIDA sites, with a 34% rate of insulin absorption and a need for a doubled insulin dose compared to injection into normal sites [9]. Suboptimal blood glucose control often requires an increased insulin dosage, which may lead to a patient’s weight gain [17] and subsequent worsening of management, especially in type 2 diabetes. The mechanism of how amyloid prevents insulin from its pharmacodynamic effect remains unclear. Nilsson proposed three possibilities: 1. injected insulin may not be able to go through the deposit and reach the blood; 2. preformed amyloid fibrils convert monomeric insulin into insoluble amyloid fibrils; and 3. insulin-derived amyloidosis is further modulated by an insulin-degrading enzyme which removes the injected insulin [22]. Amyloid is a protein that basically cannot be removed from the body—from this point of view, the second possibility seems plausible, with unknown rationale, however, as to why the iatrogenic insulin turns into amyloid in some people.

In addition to LIDA, the clinical differential diagnosis of local long-term insulin application complications includes local allergic reactions manifesting in erythema, pruritus, painless swelling, abscess formation, scarring (particularly in insulin pump use), lipoatrophy/dystrophy, and lipohypertrophy [25]. Lipohypertrophy presents as a soft cutaneous mass at the site of insulin injections, and it is commonly compared to LIDA; it is estimated that 50% of type 1 diabetic patients may develop lipohypertrophy over the course of their insulin therapy [26]. Lipohypertrophy is caused by the lipogenic and anabolic properties of insulin. Histologically, it is characterized by lobular proliferation of matured adipose tissue with focal fibrosis and edematous changes around the insulin injection site [1]. Similar to LIDA, advanced lipohypertrophy may lead to decreased insulin absorption due to fibrosis and hypovascularized tissue [27]. The lesion can be resolved by switching of the insulin injection site and does not usually require surgical intervention [1].

In contrast to this, lipoatrophy is considered to have an immunological basis, predisposed by the lipolytic components of certain insulins [25]. Similarly to LIDA and lipohypertrophy, the absorption of insulin from lipoatrophic areas leads to frequent difficulties in achieving ideal blood glucose control. Lipodystrophy is seen twice as commonly with medium- or long-acting insulin compared to regular short-acting insulin because it stay longer at the injection site and provides a source for local antigens [26].

All three lesions, LIDA, lipohypertrophy, and lipodystrophy, are clinically characterized as a painless subcutaneous mass at the insulin injection site, related to poor glycemic control and the need for a higher insulin dosage. All of them may be prevented by patient education to regularly rotate the injection sites. Contrary to the latter lesions, LIDA, like other forms of amyloidosis, cannot be resolved spontaneously, and it requires surgical excision [1]. In contrast to the palpation softness in lipohypertrophy and lipodystrophy, LIDA is usually notably firm. Clinically, LIDA is probably frequently overlooked and undertreated; therefore, firm local complaints following insulin injection should be considered to undergo a biopsy, whereas histology is the only method leading to accurate LIDA diagnosis. Treatment of LIDA involves surgical excision of the lesion and avoiding insulin injection at the amyloidosis site [28]. There may be a risk of hypoglycemia in patients when they immediately switch the injection site from a spot with cutaneous amyloidosis to an unaffected site. To prevent this, patients should carefully notice blood glucose levels after switching injection areas and consider regulating the dose of insulin or antidiabetic medication.

On the other hand, based on clinical and histological examination without immunohistochemistry, LIDA may be confused with local involvement in systemic amyloidosis, as a native subcutaneous fat biopsy is routinely used in amyloid diagnostics. Additionally, diabetic nephropathy manifesting with proteinuria and kidney function failure may be erroneously attributed to systemic amyloidosis with kidney involvement. Therefore, in any incidental or targeted subcutaneous amyloid finding, immunohistochemistry or immunofluorescence aimed at exact amyloid classification should be performed. Alternatively, amyloid typing may be performed by laser microdissection from FFPE tissue and mass spectrometry-based proteomic analysis [29], as several amyloid types including insulin have been described [30]. In the case of amyloidosis verified in the tissue sample, the localized amyloidosis may be distinguished from systemic involvement by ^18^F-fluorodeoxyglucose positron emission tomography/computed tomography [31].

LIDA diagnostics and treatment are limited by its rarity as there are mainly case reports or small case series available in the literature. Small datasets do not allow for extensive statistical analysis and therapeutic guideline establishment.

## 4. Conclusions

LIDA represents a rare but clinically significant consequence of long-term insulin injection. Clinical suspicion and knowledge of the lesion play crucial roles in the accurate diagnosis of LIDA. Diagnosis is based on the histological identification of amyloid fibrils and anti-insulin immunoreactivity. Surgical excision and change in the insulin application site may significantly improve hyperglycemia compensation.

## Figures and Tables

**Figure 1 diagnostics-13-02415-f001:**
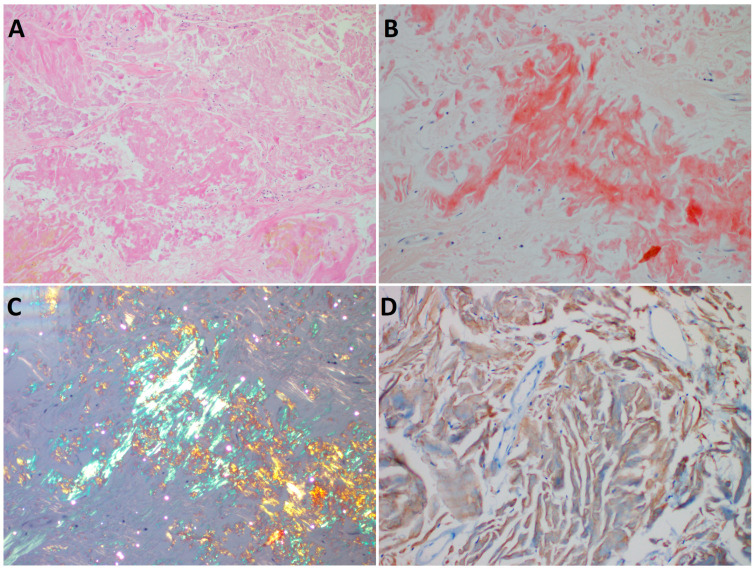
(**A**): Hematoxylin–eosin-stained microscopical image (200×) showing abundant amorphous eosinophilic extracellular deposits in the dermis and in the subcutaneous fat tissue, with occasional fibrosis, without any inflammatory cell infiltration. (**B**): The deposits display positive Congo red staining. (**C**): Apple-green birefringence in polarized light. (**D**): The immunohistochemical examination shows slight but unequivocal insulin positivity (BioSB, USA, clone BSB42, 1:1000).

## Data Availability

Not applicable.

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
