# Peer review of "Localized Insulin-Derived Amyloidosis in Diabetes Mellitus Type 1 Patient: A Case Report"

_diagnostics, 2023, doi:10.3390/diagnostics13142415_

Round 1

Reviewer 1 Report

Small subsections may be shown

Author Response

ANSWER: Dear reviewer, thank you for your positive reaction. In line with your comment, we put in the revised manuscript subsections: Introduction, Case description, Discussion, Conclusion.

Reviewer 2 Report

Dear Authors,

Manuscript ID diagnostics-2456384   It was interesting to review this case report. I have a few suggestions below:

Authors said "The lesion can be resolved by cessation of insulin injection.." (line no: 133)
what you suggest as an alternative during cessation

What about the applications of Mass spectrometry based proteomics in diagnosis like amyloid typing by LMD-MS/MS as well as FDG-PET CT.

I think,
There may be a risk of hypoglycaemia in patients when they immediately switch injection site from a spot with cutaneous amyloidosis to an unaffected site.

So the following lines can be be included as an advise in your report:

Patients should carefully notice blood glucose level after switching injection area and consider regulating the dose of insulin or antidiabetic medication to prevent hypoglycaemia.

There were a few formatting errors, kindly correct them.

Include the limitations of your study.

Also, include sub-section headings like case presentation, discussion, limitation of the study, conclusion, therefore report will become clearer.

Author Response

Dear reviewer, thank you for the valuable comments.

As you correctly suggest, insulin cessation is not feasible in patients with insulinodependent diabetes. We ammended the discussion about lipohypertrophy as follows:

"The lesion can be resolved by switching of insulin injection site and usually does not require surgical intervention."

Based on your comment, we added in the discussion brief section mentioning LMD-MS/MS and FDG-PET CT: 

"Therefore, in any incidental or targeted subcutaneous amyloid finding, immunohistochemistry or immunofluorescence aiming to exact amyloid classification should be performed. Alternatively, amyloid typing may be performed by laser microdissection from FFPE tissue and mass spectrometry-based proteomic analysis [29], several amyloid types including insulin have been described [30]. In case of amyloidosis verified in tissue sample, the localized amyloidosis may be distinguished from the systemic involvement by 18F-fluorodeoxyglucose positron emission tomography/computed tomography [31]."

The LIDA treatment has been briefly discussed in the revised manuscript including your kind suggestions:

"Treatment of LIDA involves surgical excision of the lesion and avoiding insulin injection at amyloidosis site [28]. There may be a risk of hypoglycemia in patients when they immediately switch injection site from a spot with cutaneous amyloidosis to an unaffected site. To prevent this, patients should carefully notice blood glucose level after switching injection area and consider regulating the dose of insulin or antidiabetic medication."

We included into a the discussion a short paragraph describing limitation of the study:

"LIDA diagnostics and treatment are limited by its rarity as there are mainly case reports or small case series available in the literature. Small datasets do not allow an extensive statistical analysis and therapeutic guideline establishment."

I found some typing errors - line 20 "caUse", line 54 "compensasion haS been poor" - these have been amended.

In line with your suggestion, the manuscript is newly subsectioned into Introduction, Case description, Discussion, Conlusion

Reviewer 3 Report

Comments:

1.      The article should be divided into four parts: Introduction, Case Report, Discussion and Conclusion

2.      Please discuss the treatment of Localized insulin-derived amyloidosis and add conclusion. 

Author Response

Dear reviewer, thank you for your valuable comments.

ANSWER:

  1. In line with your comment, we put in the revised manuscript subsections: Introduction, Case description, Discussion, Conclusion.
  2. We broadened the discussion with a brief treatment description as follows: "Treatment of LIDA involves surgical excision of the lesion and avoiding insulin injection at amyloidosis site [28]. There may be a risk of hypoglycemia in patients when they immediately switch injection site from a spot with cutaneous amyloidosis to an unaffected site. To prevent this, patients should carefully notice blood glucose level after switching injection area and consider regulating the dose of insulin or antidiabetic medication."

Round 2

Reviewer 3 Report

Thanks for edition. It is well done.